

# Estimation of different data compositions for early-season crop type classification

Pengyu Hao[1,2,3], Mingquan Wu[2], Zheng Niu[2], Li Wang[2] and Yulin Zhan[2]

[1] Key Laboratory of Agricultural Remote Sensing, Ministry of Agriculture, Chinese Academy of Agricultural Sciences, China. (AGRIRS)/Institute of Agricultural Resources and Regional Planning, Beiijng, China
[2] The State Key Laboratory of Remote Sensing Science, Institute of Remote Sensing and Digital Earth, Chinese Academy of Sciences, Beijing, China
[3] Key Laboratory for Geo-Environmental Monitoring of Coastal Zone of the National Administration of Surveying, Mapping and GeoInformation & Shenzhen Key Laboratory of Spatial Smart Sensing and Services, Shenzhen University, Shenzhen, China

Corresponding authors
Pengyu Hao, haopy8296@163.com
Zheng Niu, niuzheng@radi.ac.cn

## ABSTRACT

Timely and accurate crop type distribution maps are an important inputs for crop yield estimation and production forecasting as multi-temporal images can observe phenological differences among crops. Therefore, time series remote sensing data are essential for crop type mapping, and image composition has commonly been used to improve the quality of the image time series. However, the optimal composition period is unclear as long composition periods (such as compositions lasting half a year) are less informative and short composition periods lead to information redundancy and missing pixels. In this study, we initially acquired daily 30 m Normalized Difference Vegetation Index (NDVI) time series by fusing MODIS, Landsat, Gaofen and Huanjing (HJ) NDVI, and then composited the NDVI time series using four strategies (daily, 8-day, 16-day, and 32-day). We used Random Forest to identify crop types and evaluated the classification performances of the NDVI time series generated from four composition strategies in two studies regions from Xinjiang, China. Results indicated that crop classification performance improved as crop separabilities and classification accuracies increased, and classification uncertainties dropped in the green-up stage of the crops. When using daily NDVI time series, overall accuracies saturated at 113-day and 116-day in Bole and Luntai, and the saturated overall accuracies (OAs) were 86.13% and 91.89%, respectively. Cotton could be identified 40~60 days and 35~45 days earlier than the harvest in Bole and Luntai when using daily, 8-day and 16-day composition NDVI time series since both producer's accuracies (PAs) and user's accuracies (UAs) were higher than 85%. Among the four compositions, the daily NDVI time series generated the highest classification accuracies. Although the 8-day, 16-day and 32-day compositions had similar saturated overall accuracies (around 85% in Bole and 83% in Luntai), the 8-day and 16-day compositions achieved these accuracies around 155-day in Bole and 133-day in Luntai, which were earlier than the 32-day composition (170-day in both Bole and Luntai). Therefore, when the daily NDVI time series cannot be acquired, the 16-day composition is recommended in this study.

## INTRODUCTION

Multi-temporal remote sensing data, particularly vegetation indices (VI) time series, can describe crop conditions during the crop growing season and have been widely used to classify crop types (*Boryan et al., 2011*; *Waldhoff, Lussem & Bareth, 2017*; *Wardlow & Egbert, 2008*). Most previous studies have employed remote sensing images of the entire growing season to generate crop type distribution maps (*Howard & Wylie, 2014*; *Zhang, Feng & Yao, 2014*), and the crop maps are generally acquired after the crop harvest. However, timeline is the first factor when considering a crop type map because an early classification result might benefit decision-makers and the private sector by helping growth monitor, forecasting crop yield and qualifying crop drought (*Gallego et al., 2008*; *Skakun et al., 2017*).

Considerable researches have been conducted on early-season crop type mapping (*Azar et al., 2016*; *Vaudour, Noirot-Cosson & Membrive, 2015*; *Villa et al., 2015*), and there are two vital factors that contribute to early crop identification: the crop calendar and the remote sensing imagery characteristics. Previous early crop identification studies have shown that if crops are separable, high temporal frequency data such as Moderate Resolution Imaging Spectroradiometer (MODIS) can identify crops with a short image time series (*Hao et al., 2015*; *Zhou, Zhang & Townley-Smith, 2013*). The spatial resolution of high temporal density data, however, are relatively coarse (*Verbeiren et al., 2008*). Thus, mixed pixels may lead to serious misclassification in a heterogeneous landscape (*Hao, Wang & Niu, 2015b*). At finer spatial resolution, the possibility of obtaining imagery with dense temporal resolution is low. For example, Landsat ETM+ cannot provide cloud-free image in each season globally, especially during autumn and winter (*Ju & Roy, 2008*). Although Landsat and Huanjing (HJ) images have been used to improve the temporal resolution of image time series (*Hao et al., 2014*); in addition, Sentinel-2 satellite may provide optical images with higher spatial and temporal resolutions (10m and five-day) (*European Space Agency, 2016*), but the image time series still contain "missing values" due to cloud cover (*Hao, Wang & Niu, 2015a*).

Most classifiers cannot handle "missing values" within the image time series, but image time series at a good spatial resolution. such as 30 m, are generally irregular because of low satellite revisit frequencies and cloud cover. One commonly used method used to reduce the "missing value" pixels is image composition (*Xiong et al., 2017*). However, the optimal composition period for cropland identification is not clear as short composition periods might describe dynamic land cover changes accurately, but missing pixels caused by cloud cover are also more likely to be included in the short composited image time series (*Van Leeuwen, Huete & Laing, 1999*), and density time series generated from short compositions also lead to information redundancy (*Low et al., 2013*; *Wardlow & Egbert, 2010*). Therefore, the composition period should be a balance between the number of missing pixels and the density of image time series.

To estimate the optimal composition period for crop type mapping at 30 m resolution, an image time series with a high temporal density is firstly generated by fusing coarse and medium spatial resolution data (*Wu et al., 2015a*). *Gao et al. (2006)* introduced the

Spatial and Temporal Adaptive Reflectance Fusion Model (STARFM) to blend MODIS and Landsat data. Some other data fusion methods are based on the assumption that MODIS pixels are a linear combination of each contributing land cover class (*Maselli, Gilabert & Conese, 1998*; *Zhukov et al., 1999*), but this is not the real-case scenario (*Wu et al., 2015a*). *Wu et al. (2012)* assumed that pixels of the same land cover have similar temporal variation characteristics and proposed a spatial and temporal data fusion approach (STDFA). This method has the potential to synthesize daily imagery and products such as land surface temperature (LST) and leaf area index (LAI) at a medium resolution (*Wu et al., 2015b*; *Wu et al., 2015c*).

The objectives of the study were to: (1) generate NDVI time series of four different composition periods (daily, 8-day, 16-day and 32-day) from daily NDVI time series, and evaluate classification performances of the four image compositions, and (2) test and analyze the potential of early-season crop identification using the NDVI time series of the four compositions.

## STUDY REGIONS AND DATA SETS

### Study regions

Xinjiang is the most important cotton-growing region in China, contributing half of the national cotton yield (*Wang et al., 2014*; *Xu, Xinxiang & Xiuju, 2007*). Early crop identification could help manage crop cultivation, particularly in terms of the cotton monitor. In this study, two representative study regions containing major crops in Xinjiang were selected.

The two study regions are located in Xinjiang Province, China (Fig. 1). The first study region is Bole County (44°20′–45°23′N, 80°40′–82°42′E), located in the northwest part of Xinjiang. The region has a temperate, continental climate characterized by dryness and drought. The annual average temperature and rainfall are 7.0 °C and 202 mm. Bole County is a representative region in north Xinjiang because this county contains the major crops such as cotton and grapes. The crop calendar of the crops in this study region is shown in Fig. 2. Cotton and maize are sown in the middle of April and develop between May and August. Cotton is harvested in the middle of August and spring maize is harvested in September. Grapes blossom in April and develop between April and June. Multiple grapes varieties lead to long harvest period of grapes harvest (Fig. 2). Watermelon is sown in early May, develops between May and August and is then harvested in late August.

The second study region is Luntai County (41°39′–41°56′N, 84°00′–84°21′E), located at the Bayinguoleng Mongolian Autonomous Prefecture between Southern Tianshan and the north of the Tarim Basin. It has a warm, temperate, continental, arid climate, and the annual average temperature and rainfall are 10.9 °C and 52 mm. The main crops of Luntai are cotton, maize, melons and winter wheat, and the crop calendar of these crops is shown in Fig. 2. Winter wheat is sown in early October, develops between October and the following May and matures in early June. After the winter wheat is harvested, summer maize is sown, which develop over July and August, and is harvest in September. Therefore, these fields have a winter wheat-summer maize rotation, and we defined them as "wheat-maize" in this study. Cotton, spring maize and melons are sown in April, melon develops between

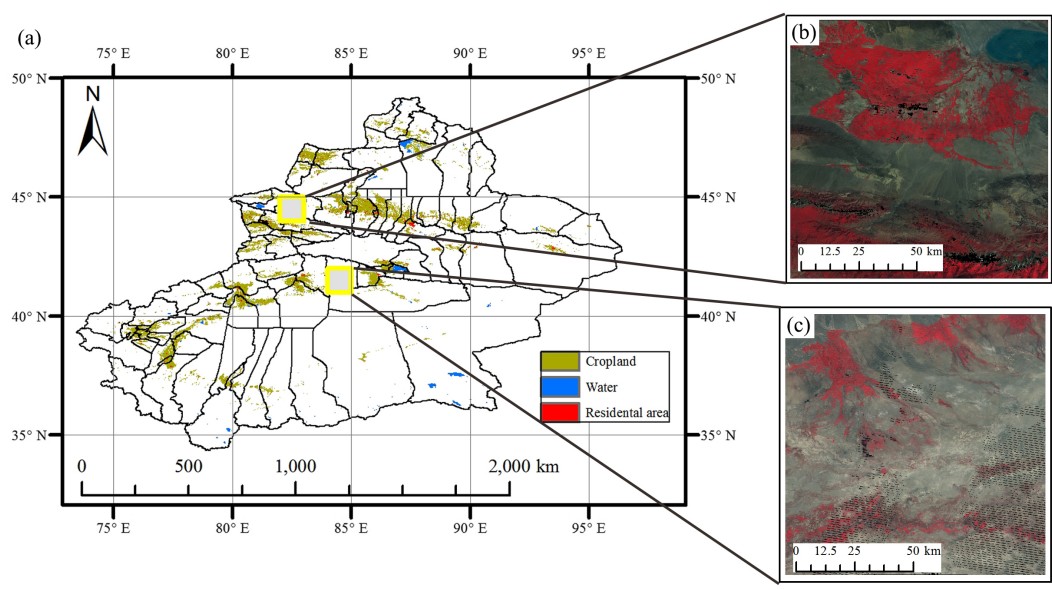

**Figure 1   The location of Study regions in Xinjiang.** (A) Locations of study regions; (B) false color image of Bole; (C) false color image of Luntai.

(a)   Bole

| Crop | Stage | Jan | Feb | Mar | Apr | May | Jun | Jul | Aug | Sep | Oct | Nov | Dec |
|------|-------|-----|-----|-----|-----|-----|-----|-----|-----|-----|-----|-----|-----|
| | | I II III | I II III | I II III | I II III | I II III | I II III | I II III | I II III | I II III | I II III | I II III | I II III |
| Cotton | Planting | | | | ■ | | | | | | | | |
| | Developing | | | | ■ | ■ | ■ | ■ | ■ | | | | |
| | Harvesting | | | | | | | | | ■ | ■ | | |
| Grape | Planting | | | | ■ | | | | | | | | |
| | Developing | | | | ■ | ■ | ■ | | | | | | |
| | Harvesting | | | | | | | ■ | ■ | ■ | ■ | | |
| Watermelon | Planting | | | | ■ | | | | | | | | |
| | Developing | | | | ■ | ■ | ■ | | | | | | |
| | Harvesting | | | | | | | ■ | ■ | ■ | | | |
| Spring Maize | Planting | | | | ■ | | | | | | | | |
| | Developing | | | | | ■ | ■ | ■ | ■ | | | | |
| | Harvesting | | | | | | | | | ■ | | | |

(b)   Luntai

| Crop | Stage | Jan | Feb | Mar | Apr | May | Jun | Jul | Aug | Sep | Oct | Nov | Dec |
|------|-------|-----|-----|-----|-----|-----|-----|-----|-----|-----|-----|-----|-----|
| | | I II III | I II III | I II III | I II III | I II III | I II III | I II III | I II III | I II III | I II III | I II III | I II III |
| Winter Wheat | Planting | | | | | | | | | | ■ | | |
| | Developing | ■ | ■ | ■ | ■ | ■ | ■ | | | | ■ | ■ | ■ |
| | Harvesting | | | | | | ■ | | | | | | |
| Spring Maize | Planting | | | | ■ | | | | | | | | |
| | Developing | | | | ■ | ■ | ■ | ■ | | | | | |
| | Harvesting | | | | | | | | ■ | | | | |
| Summer Maize | Planting | | | | | | ■ | | | | | | |
| | Developing | | | | | | ■ | ■ | ■ | | | | |
| | Harvesting | | | | | | | | | ■ | | | |
| Cotton | Planting | | | | ■ | | | | | | | | |
| | Developing | | | | ■ | ■ | ■ | ■ | ■ | ■ | | | |
| | Harvesting | | | | | | | | | ■ | ■ | | |
| Melon | Planting | | | | ■ | | | | | | | | |
| | Developing | | | | ■ | ■ | ■ | | | | | | |
| | Harvesting | | | | | | | ■ | | | | | |

**Figure 2   Crop calendar of the major crops in this study.** (A) Crop calendar in Bole; (B) Crop calendar in Luntai.

April and July and is harvest in early August, cotton develops between May and August and is harvested in September and October, spring maize develops between May and the middle of August and is harvested in late August and early September. We define spring maize as "maize" in this study. As well as the crops, there are also some orchards with apple and pear trees in Luntai. These trees blossom in April and the fruits become ripe in late August/September.

## Density NDVI time series

The daily NDVI time series at the 30 m resolution were obtained by fusing cloud-free MODIS, Landat, HuanJing (HJ) and GaoFen (GF) NDVI during Day 97~320 for Bole in 2011 and Day 93~315 Luntai in 2013 (*Wu et al., 2015d*). Landsat, HJ and GF data have medium spatial resolution (Landsat, HJ 30 m and GF-WFV 16 m). All these data were surface reflectance and the data from Bole were georeferenced with the UTM WGS 84, zone 44N. The data from Luntai were georeferenced with the UTM WGS 84, zone 45N. The HJ and GF data were registered to the TM images, achieving an RMSE of less than 0.3 pixels using a second order polynomial transformation and bi-linear resampling, and the spatial resolution of the data were resampled to 30 m. As these three medium resolution sensor systems (Landsat, GF and HJ) have differences in parameters, bandwidth, acquisition time and spectral response (*Gao et al., 2006*), the Landsat and GF NDVI were calibrated to HJ NDVI because HJ acquired the most cloud-free observations among the three sensor systems. The MOD09GA land surface reflectance product was employed in this study, and the NDVI was then calculated. Next, we used the STDFA to fuse the MODIS NDVI and HJ, GF NDVI and generated the daily NDVI time series. The fusion method and the validation of the fusion NDVI time series were introduced by *Wu et al. (2015d)* in detail. For each date, if we were able to acquire data in one of the four sensor systems, the NDVI at 30 m was recorded and vice versa. In this study, we tried to estimate the potential of early season crop type classification and estimate the performance of the four image time series compositions based on the daily NDVI time series at the 30 m resolution generated from the MODIS, Landsat, GF and HJ NDVI fusion.

One limitation is that NDVI profiles of some crops included some confusion, which had negative effects on classification performance. Previous studies have shown that multi-spectral bands can achieve higher classification accuracy compared with NDVI (*Waldner et al., 2015*). However, we only used NDVI in this study because the daily time series were obtained by fusing MODIS NDVI with medium-resolution NDVI data from multi-sensors. We did not fuse MODIS data and 30 m images for the multi-spectral bands because multi-spectral bands from multi-sensors have large difference due to different specific band designations and spectral response functions; while, NDVI from nultiple sensors have less difference comparing with multi-spectral bands (*Hao, Wang & Niu, 2015a*; *Hao et al., 2014*; *Wu et al., 2015a*). We have evaluated the similarity between Landsat images and HJ images, the correlation between Landsat NDVI and HJ NDVI was more similar to a 1:1 line than multi-spectral bands (*Hao et al., 2014*). In addition, the optimal features selection results showed that NDVI contributes the most to identifying crop types. Therefore, we used NDVI time series in this study, although they were not perfect (*Hao et al., 2015*).

**Table 1  Four image composition strategies.**

| Data | Daily composition | 8-day composition | 16-day composition | 32-day composition |
|---|---|---|---|---|
| 97 | $NDVI_{97}$ | $\max(NDVI_{97})$ | $\max(NDVI_{97})$ | $\max(NDVI_{97})$ |
| 98 | $NDVI_{97}, NDVI_{98}$ | $\max(NDVI_{97}{\sim}NDVI_{98})$ | $\max(NDVI_{97}{\sim}NDVI_{98})$ | $\max(NDVI_{97}{\sim}NDVI_{98})$ |
| 99 | $NDVI_{97}, NDVI_{98}, NDVI_{99}$ | $\max(NDVI_{97}{\sim}NDVI_{99})$ | $\max(NDVI_{97}{\sim}NDVI_{99})$ | $\max(NDVI_{97}{\sim}NDVI_{99})$ |
| 100 | $NDVI_{97}, NDVI_{98}, \ldots NDVI_{100}$ | $\max(NDVI_{97}{\sim}NDVI_{100})$ | $\max(NDVI_{97}{\sim}NDVI_{100})$ | $\max(NDVI_{97}{\sim}NDVI_{100})$ |
| 101 | $NDVI_{97}, NDVI_{98} \ldots NDVI_{101}$ | $\max(NDVI_{97}{\sim}NDVI_{101})$ | $\max(NDVI_{97}{\sim}NDVI_{101})$ | $\max(NDVI_{97}{\sim}NDVI_{101})$ |
| 102 | $NDVI_{97}, NDVI_{98} \ldots NDVI_{102}$ | $\max(NDVI_{97}{\sim}NDVI_{102})$ | $\max(NDVI_{97}{\sim}NDVI_{102})$ | $\max(NDVI_{97}{\sim}NDVI_{102})$ |
| 103 | $NDVI_{97}, NDVI_{98} \ldots NDVI_{103}$ | $\max(NDVI_{97}{\sim}NDVI_{103})$ | $\max(NDVI_{97}{\sim}NDVI_{103})$ | $\max(NDVI_{97}{\sim}NDVI_{103})$ |
| 104 | $NDVI_{97}, NDVI_{98} \ldots NDVI_{104}$ | $\max(NDVI_{97}{\sim}NDVI_{104})$ | $\max(NDVI_{97}{\sim}NDVI_{104})$ | $\max(NDVI_{97}{\sim}NDVI_{104})$ |
| 105 | $NDVI_{97}, NDVI_{98} \ldots NDVI_{105}$ | $\max(NDVI_{97}{\sim}NDVI_{104}),$ $\max(NDVI_{105})$ | $\max(NDVI_{97}{\sim}NDVI_{105})$ | $\max(NDVI_{97}{\sim}NDVI_{105})$ |
| 106 | $NDVI_{97}, NDVI_{98} \ldots NDVI_{106}$ | $\max(NDVI_{97}{\sim}NDVI_{104}),$ $\max(NDVI_{105}, NDVI_{106})$ | $\max(NDVI_{97}{\sim}NDVI_{106})$ | $\max(NDVI_{97}{\sim}NDVI_{106})$ |
| 107 | $NDVI_{97}, NDVI_{98} \ldots NDVI_{107}$ | $\max(NDVI_{97}{\sim}NDVI_{104}),$ $\max(NDVI_{105}, NDVI_{107})$ | $\max(NDVI_{97}{\sim}NDVI_{107})$ | $\max(NDVI_{97}{\sim}NDVI_{107})$ |
| 108 | $NDVI_{97}, NDVI_{98} \ldots NDVI_{108}$ | $\max(NDVI_{97}{\sim}NDVI_{104}),$ $\max(NDVI_{105}, NDVI_{108})$ | $\max(NDVI_{97}{\sim}NDVI_{108})$ | $\max(NDVI_{97}{\sim}NDVI_{108})$ |
| 109 | $NDVI_{97}, NDVI_{98} \ldots NDVI_{109}$ | $\max(NDVI_{97}{\sim}NDVI_{104}),$ $\max(NDVI_{105}, NDVI_{109})$ | $\max(NDVI_{97}{\sim}NDVI_{109})$ | $\max(NDVI_{97}{\sim}NDVI_{109})$ |
| 110 | $NDVI_{97}, NDVI_{98} \ldots NDVI_{110}$ | $\max(NDVI_{97}{\sim}NDVI_{104}),$ $\max(NDVI_{105}, NDVI_{110})$ | $\max(NDVI_{97}{\sim}NDVI_{110})$ | $\max(NDVI_{97}{\sim}NDVI_{110})$ |
| 111 | $NDVI_{97}, NDVI_{98}, \ldots NDVI_{111}$ | $\max(NDVI_{97}{\sim}NDVI_{104}),$ $\max(NDVI_{105}, NDVI_{111})$ | $\max(NDVI_{97}{\sim}NDVI_{111})$ | $\max(NDVI_{97}{\sim}NDVI_{111})$ |
| 112 | $NDVI_{97}, NDVI_{98}, \ldots NDVI_{112}$ | $\max(NDVI_{97}{\sim}NDVI_{104}),$ $\max(NDVI_{105}, NDVI_{112})$ | $\max(NDVI_{97}{\sim}NDVI_{112})$ | $\max(NDVI_{97}{\sim}NDVI_{112})$ |
| 113 | $NDVI_{97}, NDVI_{98}, \ldots NDVI_{113}$ | $\max(NDVI_{97}{\sim}NDVI_{104}),$ $\max(NDVI_{105}, NDVI_{112}),$ $\max(NDVI_{113})$ | $\max(NDVI_{97}{\sim}NDVI_{112}),$ $\max(NDVI_{113})$ | $\max(NDVI_{97}{\sim}NDVI_{113})$ |

## Image composition strategies

There are four image composition periods in this study: daily, 8-day, 16-day and 32-day compositions (Table 1). For each composition period, the composited NDVI was the maximum NDVI of the time range. For example, the max NDVI between Day 97 and 104 were defined as the 8-day composition NDVI during that time range; then, if the end of a compositing period is not reached, for instance at Day 100, the 8-day composited NDVI was the maximum NDVI among Day 97~100.

## Ground reference data

Ground-reference data were obtained from fieldwork in August 2011 in Bole, and September 2013 in Luntai. Several 300m*300m sampling plots were surveyed across the study areas, and the crop type information was collected. The field boundaries within the plots were recorded using GPS and digitized as polygons. Authorizations to access agricultural fields were given verbally by the National Bureau of Statistics of China (NBS) Survey Office in Xinjiang and the native farmers. Next, plot polygons were extended to field polygons based on very high resolution images (Google Earth). We overlaid the field

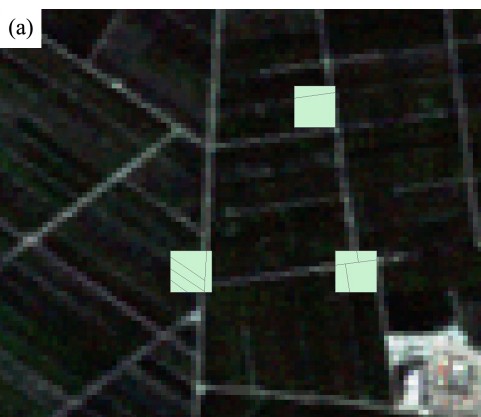
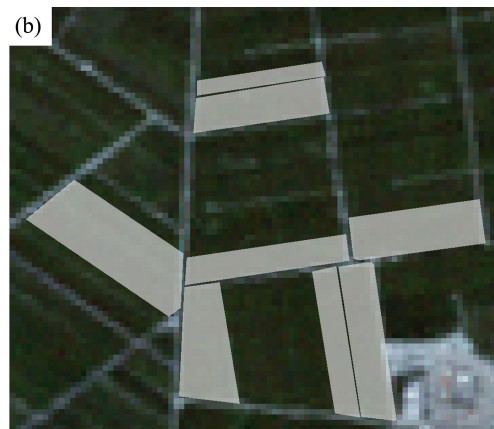
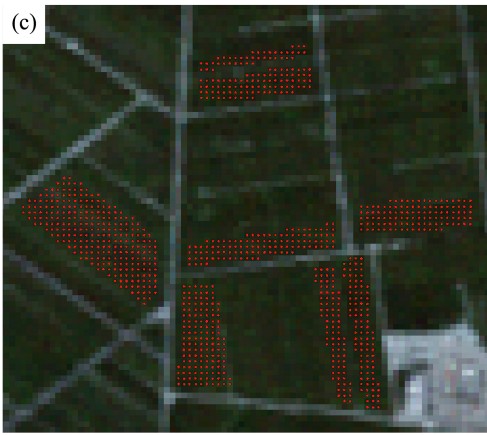

**Figure 3** **Converting the field survey plots to polygons.** (A) Three field surveyed plots; (B) the polygons extended from field plots and (C) the sample points (green points) on the Landsat image.

boundaries on Landsat images and shrank the field boundaries to ensure that all Landsat pixels enclosed in the polygons were pure crop pixels. Finally, the polygons were converted to pixel format using the TM grid (Fig. 3). Several surveyed fields (polygons) were randomly selected for training and the others were used as validation. For both training and validation polygons, a portion of samples were selected randomly from the polygons and were used as training and validation samples. The number of surveyed fields and samples is shown in Table 2.

## METHODS

### Overview of the method

A flowchart of this study is presented in Fig. 4. First, we composited the NDVI time series using four composition strategies (daily, 8-day, 16-day, and 32-day compositions). Then, both Jeffries–Matusita (JM) distance and the extension of JM distance ($J_{Bh}$) were used to calculate the crop separability. Random Forest (RF) algorithm was used to classify the crop types and classification accuracy and uncertainty were both used to evaluate classification performance. The input NDVI time series was increased from one day to

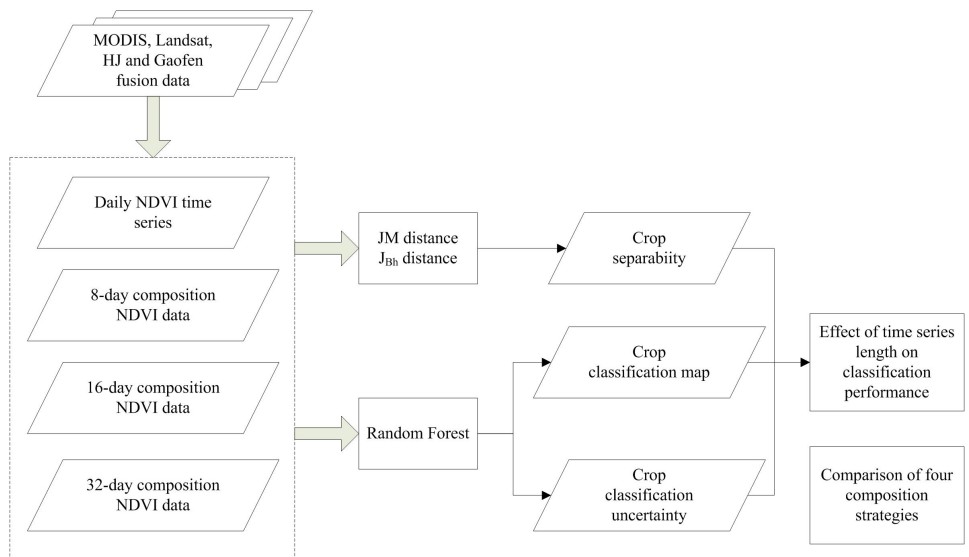

**Figure 4  Flowchart of this study.**

**Table 2  Number of surveyed fields, training and validation samples in the study area.**

| | Training | | Validation | |
|---|---|---|---|---|
| | **Polygon number** | **Sample number** | **Polygon number** | **Sample number** |
| **Bole** | | | | |
| Cotton | 54 | 1510 | 29 | 486 |
| Grape | 26 | 852 | 11 | 130 |
| Maize | 28 | 657 | 23 | 210 |
| Watermelon | 7 | 726 | 5 | 232 |
| **Luntai** | | | | |
| Cotton | 50 | 309 | 44 | 236 |
| Maize | 20 | 138 | 18 | 89 |
| Melon | 19 | 106 | 12 | 56 |
| Orchard | 23 | 118 | 25 | 91 |
| Wheat-Maize | 22 | 111 | 29 | 74 |

the entire crop growing season for each composition strategy. Finally, the potential of early season crop type mapping were evaluated using the crop separability, classification accuracy and uncertainty.

## Separability measure

We used JM distance to measure the separability of each pair-wise crop because previous research proved that JM distance have high potential to measure crop separability (*Medjahed et al., 2016*; *Murakami et al., 2001*). The JM distance between a pair of crops

could be calculated by Eq. (1):

$$\text{JM}(c_i, c_j) = \int_x \left( \sqrt{p(x|c_i)} - \sqrt{p(x|c_j)} \right)^2 dx \tag{1}$$

where $x$ is a span of VI time series values, and $c_i$ and $c_j$ are the two crop types under consideration. Then, (1) could be simplified as $\text{JM} = 2(1 - e^{-B})$, where

$$B = \frac{1}{8}(\mu_i - \mu_j)^T \left( \frac{C_i + C_j}{2} \right)^{-1} (\mu_i - \mu_j) + \frac{1}{2}\ln\left( \left| \frac{|C_i + C_j|}{2\sqrt{|C_i| \times |C_j|}} \right| \right) \tag{2}$$

$C_i$ and $C_j$ are the covariance matrices of classes $i$ and $j$, and $|C_i|$ and $|C_j|$ are the determinants of $C_i$ and $C_j$. The JM distance ranges from 0 to 2, and larger JM distance indicats higher level of separability between the two classes (*Adam & Mutanga, 2009*).

When measuring the separability of multiple classes, different classes were given different weights to account for the sample sizes. An extension of the JM distance ($J_{Bh}$) was used for this purpose (*Bruzzone, Roli & Serpico, 1995*). $J_{Bh}$ was calculated using Eq. (6) based on Bhattacharyya bounds:

$$J_{Bh} = \sum_{i=1}^{N} \sum_{j>i}^{N} \sqrt{p(w_i) \times p(w_j)} \times \text{JM}^2(i,j) \tag{3}$$

where $N$ was the number of classes and $p(w_i)$ and $p(w_j)$ were the $\alpha$ priori probabilities of classes $i$ and $j$, respectively, which were calculated using the combination of training samples in Table 1.

## Classifier

The Random Forest (RF) classifier was employed in this study. The RF model combines multiple classification trees, and the model output is determined by the majority vote of the single classification trees (*Breiman, 2001*). When training the RF model, each tree is constructed using two-thirds of the training records, and the remaining one-third of the records are used for a test classification, with an error referred to the "out-of-bag error" (OOB error). The RF could handle high dimensional data effectively and have been widely employed for land cover classification (*Immitzer, Vuolo & Atzberger, 2016*; *Rodriguez-Galiano et al., 2012*). Two free parameters of RF; the number of trees (ntree) and the number of features to split the nodes (mtry), were defined as 1000 and the square root of the total number of input features (*Loosvelt et al., 2012*), and both the crop-label and probabilistic output were obtained using the Random Forest library for R (*Breiman et al., 2013*).

## Accuracy assessment

Classification accuracy and uncertainty were both used to evaluate classification performance. The confusion-matrix-derived accuracy metrics, overall accuracy (OA), producer's accuracy (PA), user's accuracy (UA) and Kappa coefficient, were used in this study (*Congalton, 1991*). RF also provides the probabilistic output for each pixel, $(p_1(x), \ldots, p_k(x), \ldots, p_K(x), k = 1, 2, \ldots, K)$, which could be used to generate classification

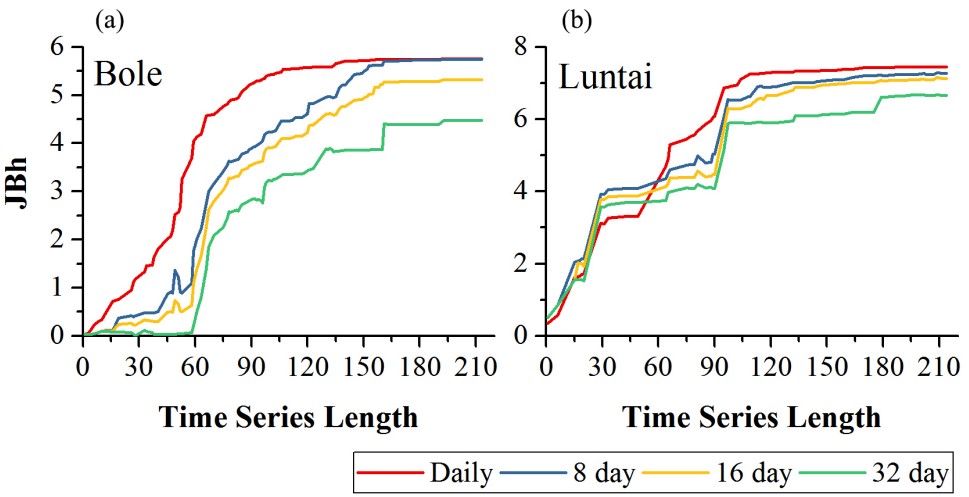

**Figure 5** **Crop separability of four image composition strategies in Bole and Luntai.** (A) Crop separability in Bole; (B) Crop separability in Luntai.

uncertainties, and we calculated the $\alpha$ quadratic entropy as classification uncertainty (*Pal & Bezdek, 1994*), (Eq. (4)):

$$H\left(p(x)\right) = \frac{1}{n \times 2^{-2\alpha}} \sum_{k=1}^{K} p_k^{\alpha}(x)\left(1 - p_k(x)\right)^{\alpha} \tag{4}$$

where $H\left(p(x)\right)$ is the $\alpha$ quadratic entropy of the vector $p(x), p_1(x), \ldots, p_K(x)$ are the probabilistic outputs, $\alpha$ is a user-defined value which ranges from 0 and 1, and $\alpha = 0.5$ is used in this study. And smaller $H\left(p(x)\right)$ means more reliable classification. One advantage of the $\alpha$ quadratic entropy is that it applies all information to the probability vector. Next, we calculated the ratio of the correctly classified pixels and wrongly classified pixels as the uncertainty ratio (Eq. (5)):

$$\text{Uncertainty Ratio} = \frac{\text{Average(Uncer}_C)}{\text{Average(Uncer}_U)} \tag{5}$$

where Average(Uncer$_C$) is the average uncertainty of all correctly identified pixels (validation samples), and Average(Uncer$_U$) is the average uncertainty of all misclassified pixels. Therefore, a lower uncertainty ratio corresponds with a better classification performance.

## RESULTS AND DISCUSSIONS

### Crop separability

Figure 5 showed the effect of time series length on crop separability for each composition strategy. For all four compositions in both study regions, $J_{Bh}$ distance increased with time series length and reached saturation points; after the saturation points, longer time series did not improve crop separability significantly. In Bole, the daily composition NDVI time series reached the saturated point the earliest (the time series length was 107-day) and

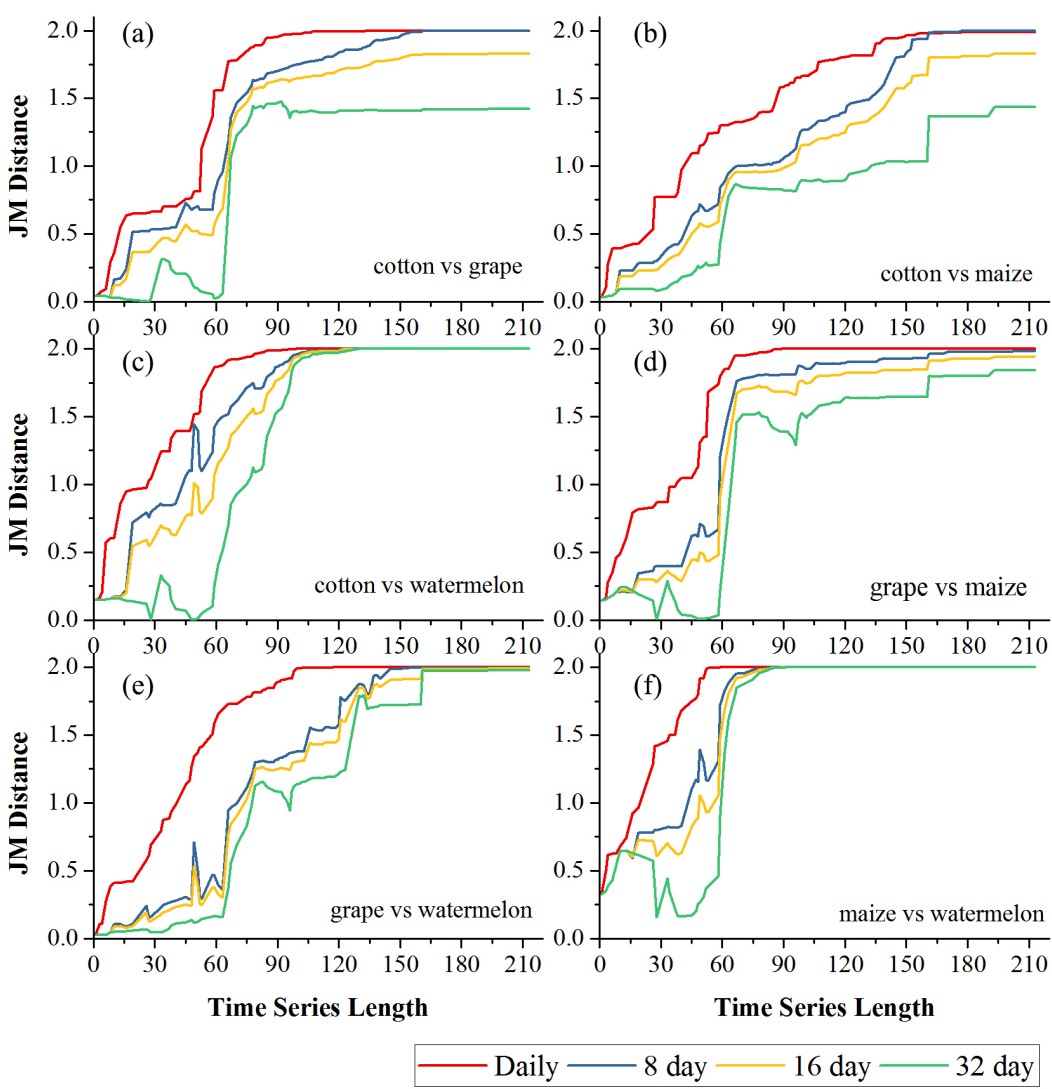

**Figure 6** **Pair-wise JM distance for different composition strategies (Bole).** (A) JM distance between cotton and grape; (B) JM distance between cotton and maize; (C) JM distance between cotton and watermelon; (D) JM distance between grape and maize; (E) JM distance between grape and watermelon; (F) JM distance between maize and watermelon.

the saturated $J_{Bh}$ was 5.53; the $J_{Bh}$ distance of the 8-day and 16-day composition NDVI time series saturated at 155-day and 158-day, and the saturated $J_{Bh}$ distances were 5.63 and 5.22, respectively. $J_{Bh}$ distance of the 32-day composition time series saturated the latest (193-day time series) and the saturated $J_{Bh}$ was 4.47. In Luntai, the $J_{Bh}$ distances of the daily, 8-day and 16-day composition time series reached saturated points at 170-day, and the saturated $J_{Bh}$ distances were 7.42, 7.21 and 7.02, respectively. The $J_{Bh}$ distance of the 32-day composition time series saturated the latest (196-day), and the saturated $J_{Bh}$ distance was 6.67.

Figures 6 and 7 showed the effect of the time series length and image composition strategies on pair-wise crop JM distances. Generally, crop separability increased with the
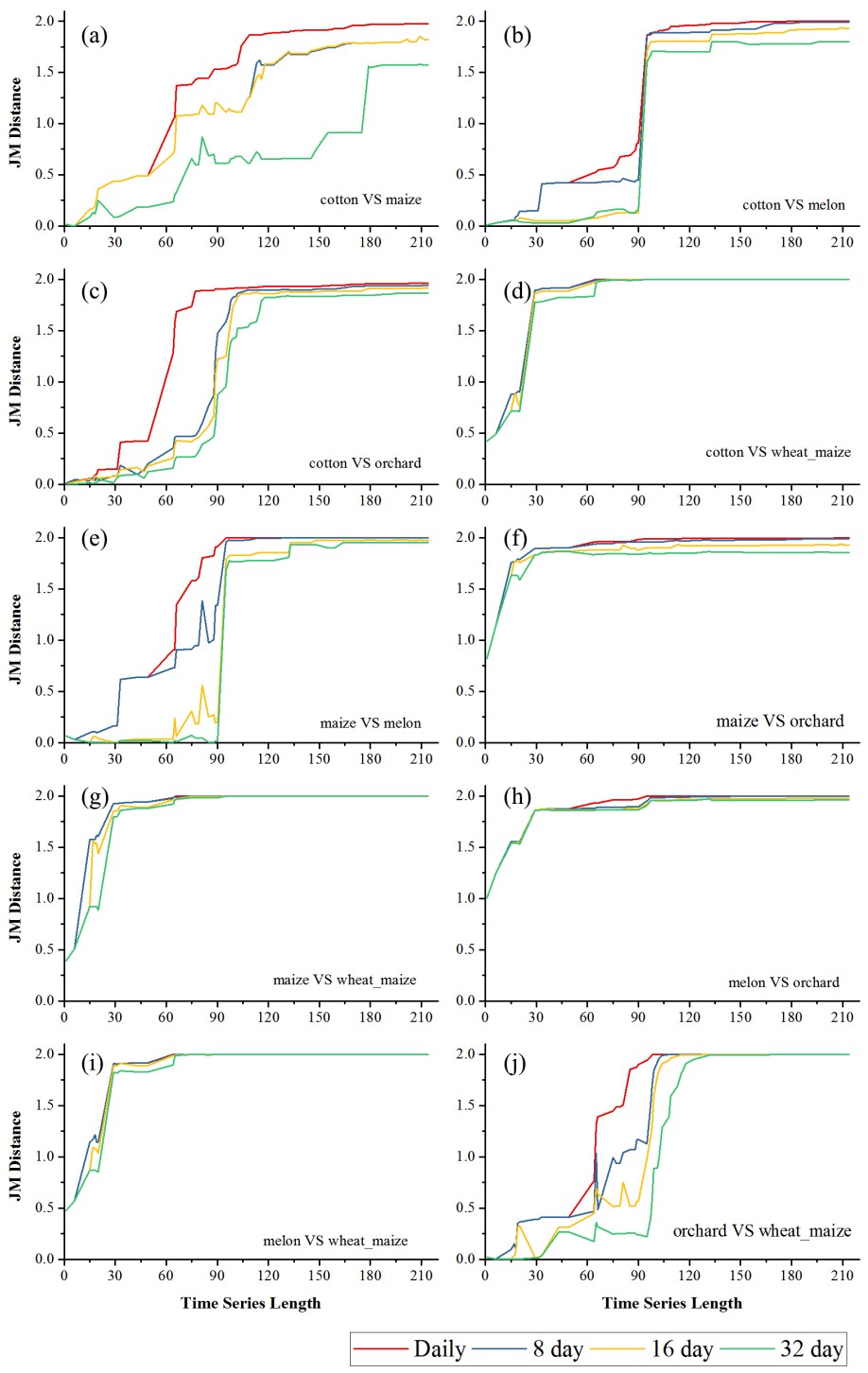

**Figure 7** **Pair-wise JM distance for different composition strategies (Luntai).** (A) JM distance between cotton and maize; (B) JM distance between cotton and melon; (C) JM distance between cotton and orchard; (D) JM distance between cotton and maize; (E) JM distance between maize and melon; (F) JM distance between maize and orchard; (G) JM distance between maize and wheat-maize; (H) JM distance between melon and orchard; (I) JM distance between melon and wheat-maize; (J) JM distance between orchard wheat-maize.

time series length and reached saturated points, and the daily time series reached saturated points the earliest and had the highest JM distance. In Bole, cotton was confused with maize and grapes. For cotton and grapes, the JM distance of the daily composition time series saturated at 88-day and the saturated JM distance was 1.95. The 8-day composition and 16-day composition had similar trends; the JM distance saturated at 134-day and 145-day, and the saturated JM distances were 1.88 and 1.80, respectively. For the 32-day composition time series, although the JM distance saturated early (at 78-day), the saturated JM distance was 1.44, which indicated that the 32-day composition time series cannot distinguish between these two crops. For cotton and maize, the JM distances of the daily, 8-day and 16-day composition time series saturated at 140-day, 153-day and 161-day. The saturated JM distances were 1.94, 1.92 and 1.80. However, the saturated JM distance of the 32-day composition time series was 1.36 at 161-day. In Luntai, wheat-maize was highly separable from the other crops in the early season (30-day). Melon crops were separable from cotton and maize at 90-day, but cotton and maize were still confusion. The JM distances of the daily, 8-day and 16-day composition time series saturated at 143-day, 170-day and 172-day. The saturated JM distances were 1.91, 1.78 and 1.77. 32-day composition had a low separability between these two crops as the saturated JM distance was 1.57 at 186-day.

## Classification accuracy

Figure 8 showed the effect of time series length on overall classification accuracy (OA) and Kappa coefficient for each composition strategy. Generally, the daily composition time series had the best OA, the 8-day and 16-day composition time series had similar performances and the 32-day composition had the lowest accuracy. In Bole, both OA and Kappa coefficient increased gradually with time series length and then saturated. The OA of the daily composition time series saturated at 113-day, and the saturated OA and Kappa coefficient were 86.13% and 0.7505. The OA of the 8-day and 16-day composition time series saturated at 153-day and 156-day, and their saturated OAs were 85.31% and 85.10%. However, the OA of the 32-day composition time series saturated at 172-day, and the saturated OA was 85.22%. In Luntai, the OA and Kappa coefficient increased at 90-day, which was consistent with the separability result that the $J_{Bh}$ distance increased at 90-day in Luntai. The OA of the daily, 8-day, 16-day and 32-day composition time series saturated at 116-day, 133-day, 132-day and 170-day. The saturated OAs were 91.89%, 83.81%, 83.95% and 82.41%.

Figures 9 and 10 showed the effect of time series length on the PAs and UAs of each composition. In Bole, the cotton PAs and UAs of the four image composition strategies had similar trends and saturated at 90-day. The saturated PAs and UAs were about 95% and 87%. The grape PAs and UAs of the daily, 8-day and 16-day image compositions showed similar trends. The PAs saturated at 90-day and the UAs saturated at 120-day. The saturated PAs and UAs were around 95% and 85%, and the PA and UA of the 32-day composition saturated late. The PA saturated at 130-day and the UA saturated at 150-day. For maize and watermelon, the daily composition time series had the highest PA and UA, and the 32-day composition data had the lowest accuracies. The 16-day and 32-day compositions performed similarly. The PA of maize was low as the saturated PA of maize was around

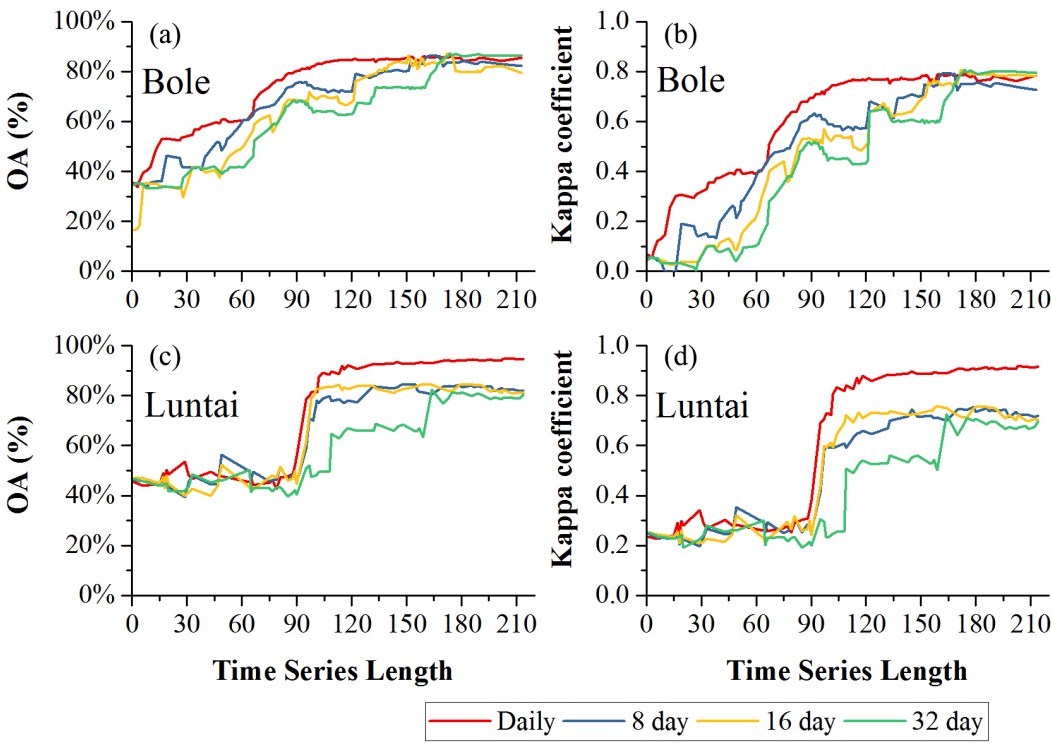

**Figure 8 Overall accuracy and Kappa coefficient in Bole and Luntai.** (A) OA in Bole; (B) Kappa coefficient in Bole; (C) OA in Luntai; (D) Kappa coeffcient in Luntai.

87%. In Luntai, the PAs and UAs of melon, orchards and wheat-maize increased at 90-day, and all four composition strategies had high accuracies (higher than 90%). Meanwhile, cotton and maize were confused, and the saturated PAs of maize were 65.3%, 65.7% and 64.6% at 123-day, 155-day and 156-day for the daily, 8-day and 16-day compositions. The maize PA of the 32-day time series was 47.2% when the entire NDVI time series were used.

Maize and grape in Bole and maize in Luntai had relatively low accuracies for all four composition strategies because of the confusion among the NDVI time series of these crops. The confusion between cotton and grape in Bole was mainly caused by the large standard deviation of grape NDVI profiles (Fig. 11), which was consistent with the large variability of grape reference NDVI time series (Hao et al., 2016). In addition, the confusion between cotton and maize in both study regions was caused by the high similarity between the cotton and maize NDVI time series (Fig. 11).

We then evaluated how early early we can identify crops (when both the PA and UA are higher than 85%, Table 3) using time series with different composition strategies. In Bole, as the cotton was harvested in late August (Fig. 2), it could be identified 60 days before the harvest using the daily NDVI time series and 40 days before the harvest using the 8-day and 16-day composition time series. In Luntai, cotton was harvested in early September as the PAs and UAs of the daily, 8-day and 16-day NDVI time series were higher than 85% at 104-day, 113-day and 115-day, and the cotton could be mapped 35~45 days before harvest.

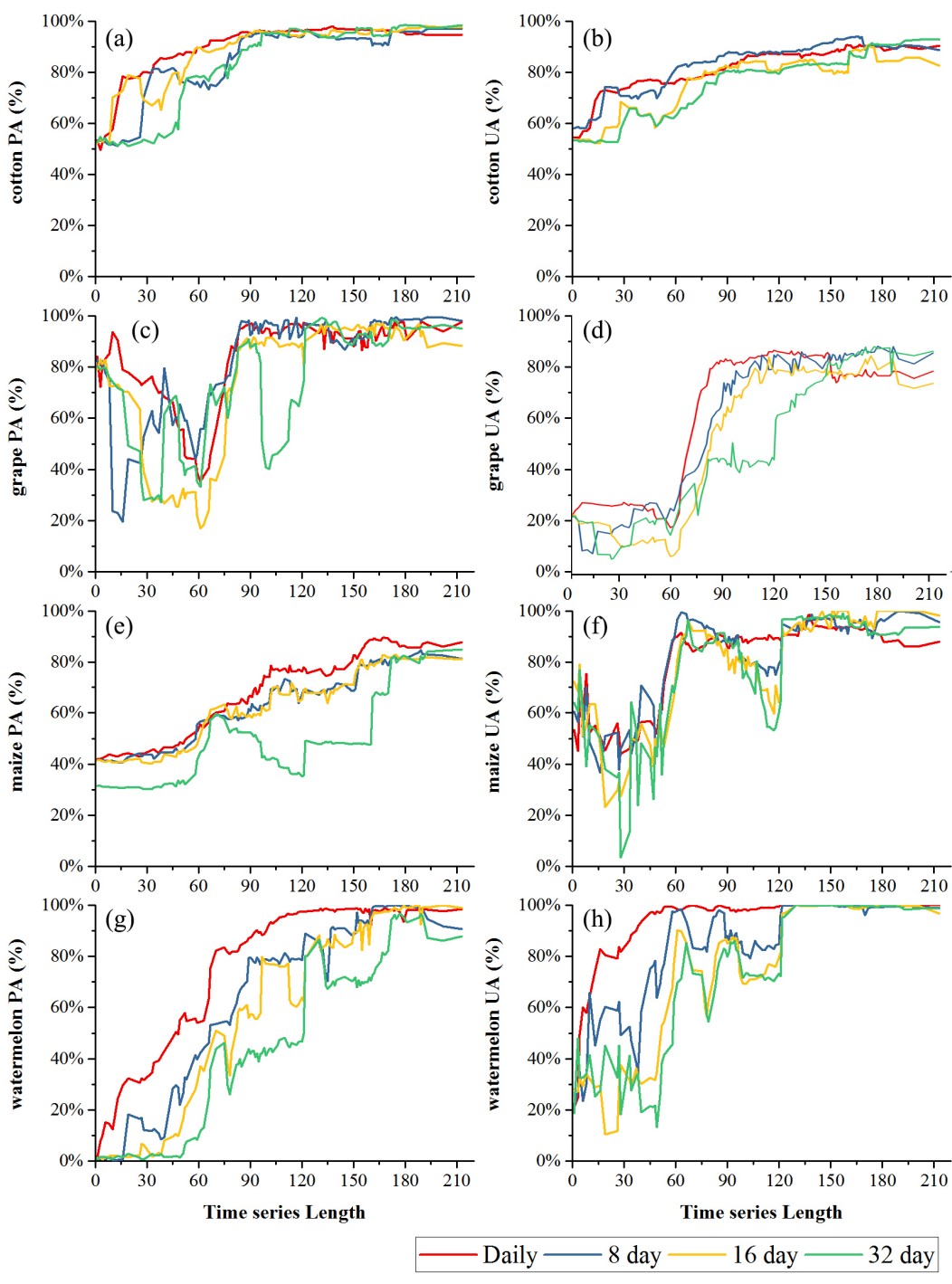

**Figure 9** **Producer's accuracies and user's accuracies in Bole.** (A) PA of cotton; (B) UA of cotton; (C) PA of grape; (D) UA of grape; (E) PA pf maize; (F) UA of maize; (G) PA of watermelon; (H) UA of water-melon.

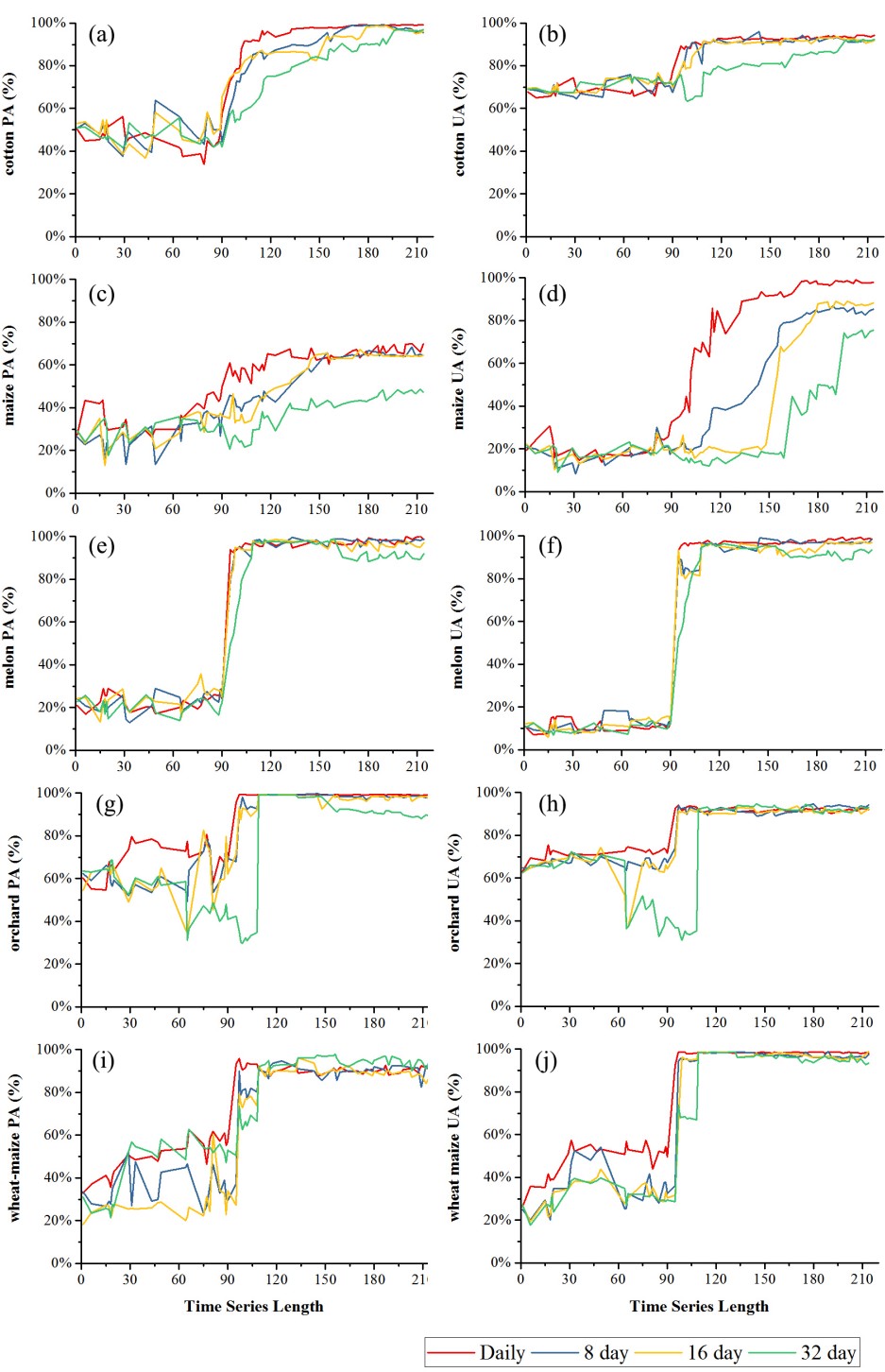

**Figure 10  Producer's accuracies and user's accuracies in Luntai.** (A) PA of cotton; (B) UA of cotton; (C) PA of maize; (D) UA of maize; (E) PA of melon; (F) UA of melon; (G) PA of orchard; (H) UA of orchard; (I) PA of wheat-maize; (J) UA of wheat-maize.

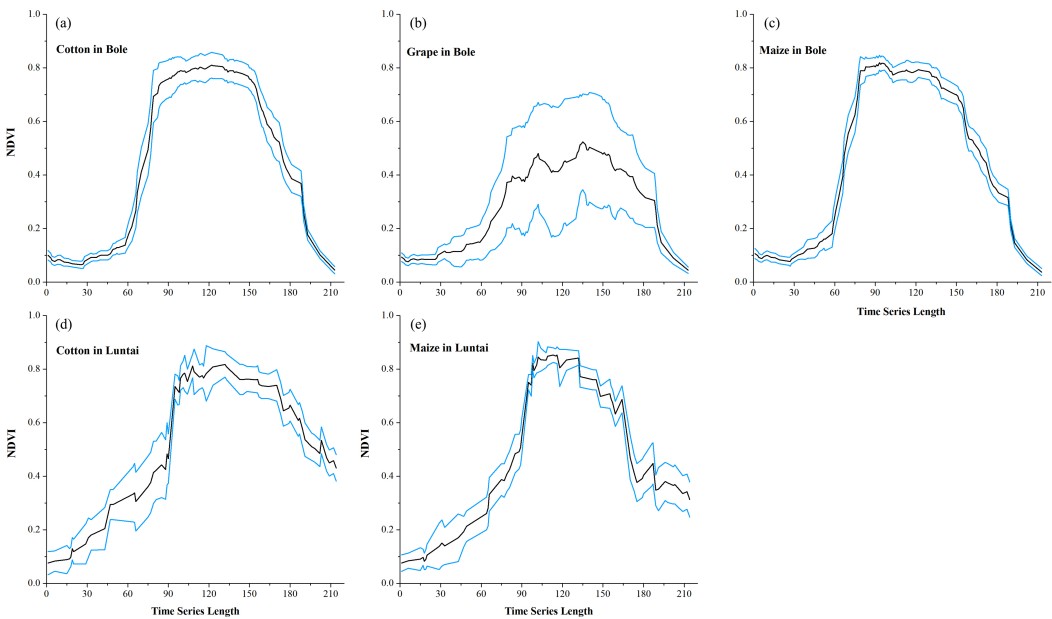

**Figure 11   Temporal NDVI time series of some confusion crops.** Black lines are the mean NDVI time series of each crop, blue lines are the Standard Deviation as Y error. (A) cotton NDVI time series in Bole; (B) grape NDVI time series in Bole; (C) maize NDVI time series in Maize; (D) cotton NDVI time series in Luntai; (E) maize NDVI time series in Luntai.

**Table 3   Shortest time series length when both the PA and UA were higher than 85%.** '-" in the table either PA or UA of the crop was lower than 85% using the corresponding composition strategy.

| | **Bole** | | | |
|---|---|---|---|---|
| | **Cotton** | **Grape** | **Maize** | **Watermelon** |
| Daily | 85 | 90 | 156 | 89 |
| 8-day | 102 | 112 | 190 | 123 |
| 16-day | 107 | 121 | 202 | 131 |
| 32-day | 162 | 153 | – | 131 |

| | **Luntai** | | | |
|---|---|---|---|---|
| | **Cotton** | **Maize** | **Melon** | **Orchard** | **Wheat-maize** |
| Daily | 104 | – | 97 | 97 | 97 |
| 8-day | 113 | – | 113 | 98 | 113 |
| 16-day | 115 | – | 113 | 98 | 113 |
| 32-day | 170 | – | 109 | 115 | 113 |

Meanwhile, as the cotton PA and UA of the 32-day composition reached 85% at 162-day and 170-day in Bole and Luntai, the 32-day composition cannot support early-season cotton identification. Accuracies of watermelon reached 85% at 89-day, 123-day, 131-day and 131-day when using the daily, 8-day, 16-day and 32-day time series. As watermelon was harvested in late August in Bole, the daily composition data was able to identify watermelon 40 days before the harvest. However, as grape was harvested between late June

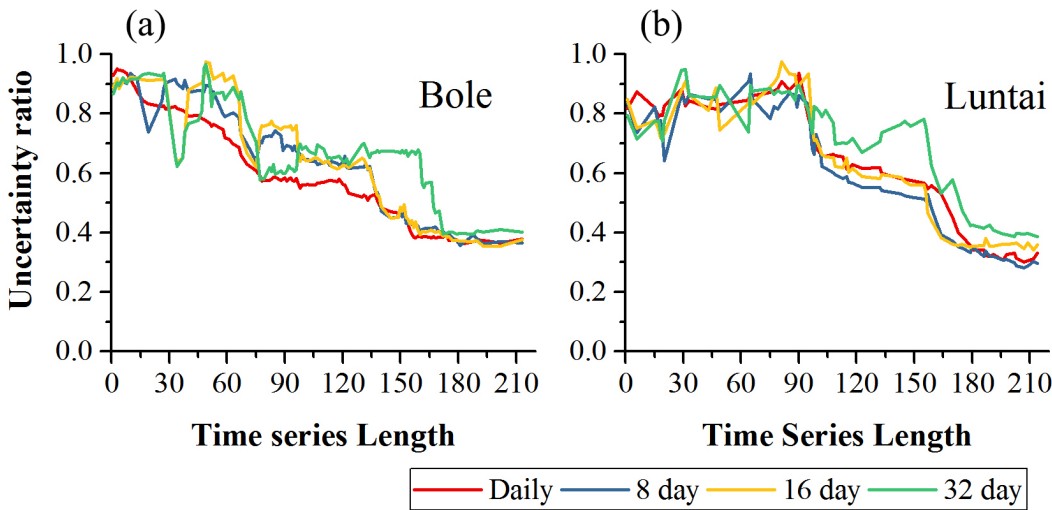

**Figure 12 Classification uncertainty ratio in Bole and Luntai.** (A) Classification uncertainty in Bole; (B) classification in Luntai.

and September in Bole and melon was harvested in early August in Luntai (Fig. 2), the early harvesting of grape in Bole and melon in Luntai cannot be correctly identified by the four compositions in the early season.

Fundamentally, dense NDVI time series can describe crop growth conditions more precisely (*Zhan et al., 2017*). For example, *Zhang, Friedl & Schaaf (2009)* found that vegetation phenology parameters could be estimated higher accuracies using denser time series. As most summer crops have different green-up speeds, dense time series have a higher probability of identifying such tiny differences. While long composition periods (such as the 32-day composition) lead to late classification saturation and low classification accuracy.

Although the daily NDVI time series achieved the best classification accuracy and the best temporal efficiency, the 30 m image time series cannot be obtained at daily frequency in the most cases. Thus, data composition should be considered in order to make full use of partly high-quality data such as the Landsat-7 SCL-off data and partly cloud-free data (*Google, 2015*; *NASA, 2015*). Among the 8-day, 16-day and 32-day compositions, the 8-day and 16-day composition data have earlier saturation data and a higher accuracy than the 32-day composition; thus, the 16-day composition strategy is recommended in this study for the identification of crop types if the daily NDVI time series cannot be acquired.

## Classification uncertainty

Figure 12 showed the classification uncertainty of the two study regions. In both study regions, the uncertainty ratio decreased with time series length. In Bole, the daily composition time series had the lowest uncertainty ratio when the time series length was shorter than 140 days. After day 140, the daily, 8-day and 16-day composition strategies had similar classification uncertainties and the saturated uncertainty was around 0.4 at 140-day. The 32-day composition had high classification uncertainty and the uncertainty

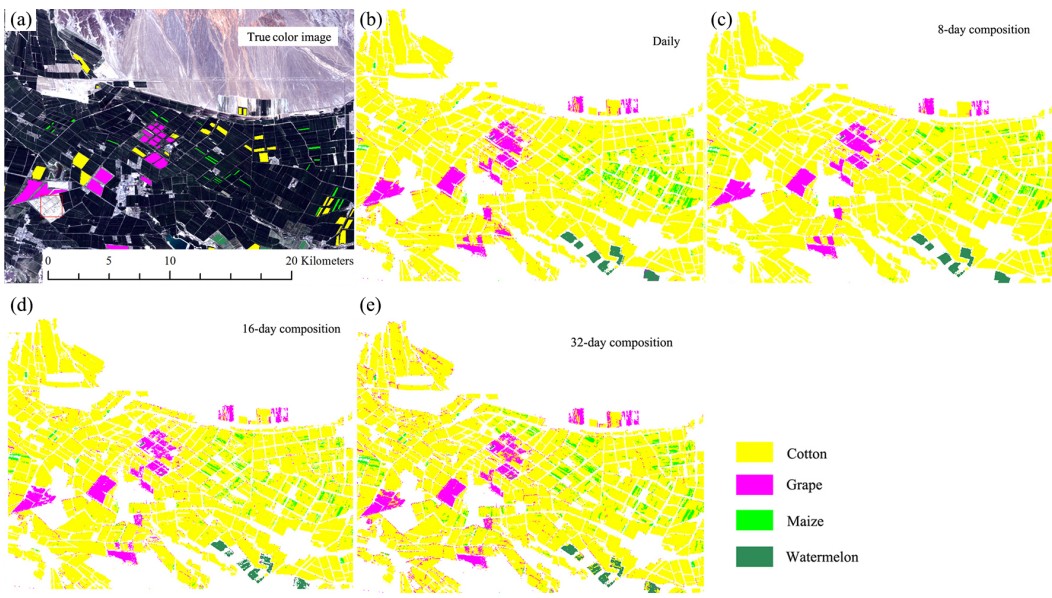

**Figure 13 Subset Crop—type mapping in Bole.** This Figure was obtained using images of the entire time series of each image composition strategies. (A) True color image and surveyed plots; (B) daily time series derived classification result; (C) 8-day time series derived classification result; (D) 16-day time series derived classification result; (E) 32-day time series derived classification result.

ratio decreased to 0.4 at 170-day. In Luntai, the daily, 8-day and 16-day time series had similar uncertainties and the uncertainty ratio dropped at 90-day and finally decreased to 0.3. The classification uncertainty of the 32-day composition time series was higher than the other compositions over the entire growing season.

## Crop distributions

Figures 13 and 14 showed that the crop distribution maps generated from the four NDVI time series composition strategies were similar. Cotton was the dominant crop in both study regions. In Bole, the crop fields were characterized by homogenous crop fields with large field sizes. Compared with crop the distribution maps generated from the daily, 8-day and 16-day composition time series, the crop map derived from the 32-day composition time series result was more speckled, in particular some cotton pixels at the field boundaries were mislabeled as grape crops. The cropland in Luntai was fragmented, so there were a large number of mixed pixels in the 30 m image time series. These mixed pixels had low NDVI because they comprised crops and the farm lanes. As melon had lower NDVI than cotton, maize and orchards, the mixed pixels were mostly misclassified as melon. As our validation samples were all pure pixels, the accuracy assessments did not describe the misclassifications of the mixed pixels in Luntai.

## CONCLUSION

This study compared the performances of daily, 8-day, 16-day and 32-day composition NDVI time series for crop types classification and reached the following conclusions:

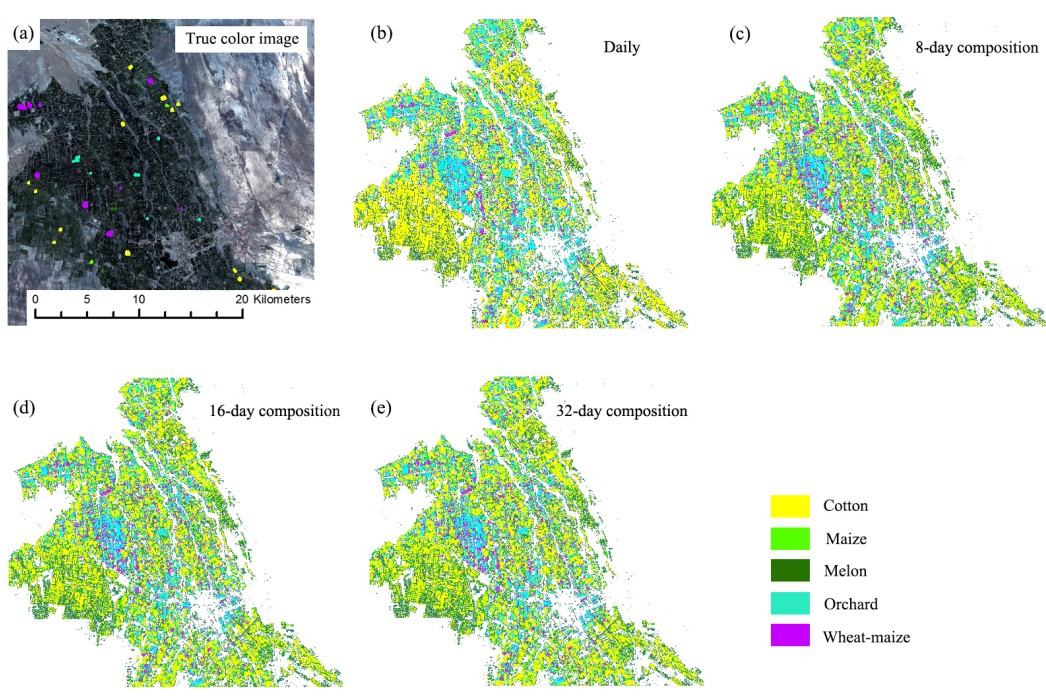

**Figure 14 Subset Crop type mapping in Luntai.** This Figure was obtained using images of the entire time series from each image composition strategy. (A) True color image and surveyed plots; (B) daily time series derived classification result; (C) 8-day time series derived classification result; (D) 16-day time series derived classification result; (E) 32-day time series derived classification result.

(1) the crops could be classified early as crop separability, classification accuracies and uncertainties saturated early. When using the daily composition NDVI time series, the OAs saturated at 113-day and 116-day, and the saturated OAs were 86.13% and 91.89% in Bole and Luntai. Longer time series could not improve the classification accuracy. (2) Cotton could be identified 40~60 days and 35~45 days before harvesting in Bole and Luntai when using the daily, 8-day and 16-day composition NDVI time series as both the PAs and UAs were higher than 85%. (3) The daily NDVI time series outperformed the other composition strategies because of the higher classification accuracies. However, if the daily NDVI time series cannot be acquired, the 16-day composition strategy was recommended in this study as the 8-day and 16-day compositions generated similar classification accuracies and outperformed the 32-day composition results. In the future, as more sensors or even geosynchronous orbit satellites (*Xu et al., 2017*) at a medium resolution become available, we would further improve the crop classification in early season.

## ACKNOWLEDGEMENTS

We thank the National Bureau of Statistics of China (NBS) Survey Office in Xinjiang and the native farmers for help us collect field data in Xinjiang.

### Funding

This work was supported by the China Postdoctoral Science Foundation funded project (No. BX201700286). The funders had no role in study design, data collection and analysis, decision to publish, or preparation of the manuscript.

### Grant Disclosures

The following grant information was disclosed by the authors:
China Postdoctoral Science Foundation: BX201700286.

### Competing Interests

The authors declare there are no competing interests.

### Author Contributions

- Pengyu Hao conceived and designed the experiments, performed the experiments, analyzed the data, contributed reagents/materials/analysis tools, prepared figures and/or tables, authored or reviewed drafts of the paper, approved the final draft.
- Mingquan Wu analyzed the data, authored or reviewed drafts of the paper, approved the final draft.
- Zheng Niu, Li Wang and Yulin Zhan authored or reviewed drafts of the paper, approved the final draft.

### Field Study Permissions

The following information was supplied relating to field study approvals (i.e., approving body and any reference numbers):

Field experiments were approved by National Bureau of Statistics of China (NBS) Survey Office in Xinjiang and the native farmers.

### Data Availability

We extracted the training samples and validation samples from the images using the samples, and attached the files.

The raw data are provided in the Supplemental File.

### Supplemental Information

Supplemental information for this article can be found online at http://dx.doi.org/10.7717/peerj.4834#supplemental-information.

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
