# Peer review of "Estimation of different data compositions for early-season crop type classification"

_PeerJ, doi:10.7717/peerj.4834_

## Round 0.1 · original submission · Major Revisions

I would urge you to pay particular regard to the comments of reviewer 3 and note the need to enhance the quality of the English (e.g. there are awkward phrases like 'time consuming' in Table 4).

Reviewer 1 ·

Basic reporting

1. Authors should conduct a thorough review for several grammatical and incorrect word usages throughout the manuscript. Examples include: Line 64 (change "An" to "A"), Line 71 (change "quantity" to "quality"), Line 130 (change "composition" to "composite period), Line 225 (change "characters" to "characteristics"), Line 297 (change "procedure's" to "producers"), Line 404 (change "phrases" to "phases") and throughout manuscript change "grape" to "grapes" unless you are using this word as an adjective.

2. Figure 3 should be changed to a Table given it is being presented in table format in the manuscript.

Experimental design

1. The authors use several moderate to high resolution remote sensing data sets (e.g., Landsat) to calculate NDVI, but do not clearly state why multiple satellite data sets are needed for the method being presented. A brief discussion of this point it needed early in the manuscript.

Validity of the findings

No comment

Additional comments

The following are recommended revision to improve the overall quality of this manuscript.

1. In the Abstract, report the classification method that was used in this work.

2. Lines 34-26: The end of this sentence does not make sense and needed to be rewritten to clearly state the point you are trying to make here.

3. Line 53: What do you mean by "short image time series"? Do you mean short historical record or short sequence of composite images during a growing season. Please better clarify.

4. Lines 57-59: Briefly discuss why Landsat ETM+ data "cannot provide" one cloud-free image in each season globally. This statement needs brief supporting discussion.

5. Line 61: State why the image data being discussed in this sentence are "still absent". Unclear in the current way it is presented.

6. Lines 64-65: There appears to be something missing from the sentence of "A simple method....". I appears it should read that a new simple method "is needed". Check the sentence and revise accordingly.

7. Lines 84-92: Suggest moving the discussion of the study area to end of this paragraph and stating the study objective to start the paragraph followed by the study area information.

8. Lines 146-147 and Figure 1: The green patterns showing the spatial distribution of the reference data are difficult to visually identify in the map of Fig 1. Change the color to make them easier to see in this graphic.

9. Line 151 and Figure 5. What are the bar presented in the line graph? This needs to be explained either in figure caption or the body of the text in the manuscript.

10. Line 285-286: Why do you think the overall accuracy decreased when the entire daily time series was used compared to shorter time series? A brief discuss of why this might have been would be useful.

11. Line 426: What type of "confusion" are you referring to here? Some discussion of the types of confusion are needed and how they negative impact the classification.

Reviewer 2 ·

Basic reporting

Paper deals with the problem of investigation of optimal satellite derived dataset for early season crop classification. Authors provide qualified literature analysis. Article structure is clear and professional. The idea to use statistical distance to investigate crop separability at satellite images is interesting.
There are some methodological issues that should be addressed.

Experimental design

A lot of experiments have been fulfilled, but the study requires additional justification.
1. First of all, the study area is very small, and the method relies on statistical distribution of different crops. This distribution could vary in different areas, so the results can be quite different for other regions. Therefore, it is unclear if the results could be extended for wider area.
2. The JM distance is calculated between a pair of probability distributions. The question is how these statistical distributions of different crops are calculated? Statistical distribution of different crops in different locations can vary.
3. What is the typical field size in the study area? The idea of Modis data downscaling using Landsat images should be justified.
4. It is unclear, how many 300x300m plots were used for training and for validation purposes.
5. What features are used for classification with random forest method? Please, clarify the structure of the model. Statement in Lines 199-200 is unclear. Please, explain, what do you mean by “the number of features to split the nodes (mtry) were defined as 1000 and the square root of the total number of input features”.
6. At Fig. 10 “Feature” should stand instead of “Feaure”.

Validity of the findings

It is unclear if the results are robust. Experiments were done only for one year for very small territory. It is not enough to provide scientific soundness of the results.
That is why conclusions could not be considered as well stated and justified.

Additional comments

The idea to investigate the optimal satellite derived dataset early season crop mapping is challenging task. Mathematical tools are interesting and promising.
I suggest to clarify the section with methodology description, to extend the study area and to fulfill the experiments for several years.

Reviewer 3 ·

Basic reporting

Language should be improved throughout the text. In particular, I recommend to use shorter and more concise sentences. For instance, line 86-92 I would suggest to split the objectives in bullet points. Line 130-131, “composition” is used four times in two rows. Another example is at Line 104: grape is a tree crop and it is not planted every year. There are a number of these examples that make the paper sometime difficult to read.
The introduction is appropriate and the background information is relevant and in line with objectives and expected results. However, the literature can be improved. There are a number of recent papers that dealt with similar problems (multi-temporal crop-type classification http://www.tandfonline.com/doi/abs/10.5721/EuJRS20164920; https://www.sciencedirect.com/science/article/pii/S0303243417300934) or used similar algorithms (e.g. Random Forest http://www.mdpi.com/2072-4292/8/3/166).
What is also missing in this work, and this is a critical point, is a reference to and use of Sentinel-2 mission data. This constellation of two satellites (S2-A and S2-B) is part of the European Copernicus programme and provides high spatial resolution data (10-20 m) with very high revisit time (10 days in China; 5-days in Europe). There are also tools (http://www.esa-sen2agri.org/) developed by the European Space Agency that support the crop type classification using multi-temporal Sentinel-2 and Landsat-8 data and this should be mentioned in the paper.
The number of figures could be reduced. The quality and the caption must be improved. For instance: Figure 1 should have a reference grid, coordinates and scale for both the region of interest and the study site. Figure 3 should be a table with column names. In the workflow in Figure 6 the arrows should be eliminated (e.g. “Separability saturation points…” is not an input to the “Classification accuracy”). In Figure 10, it is not clear what the Feature Number (x-axis) represents. Describe it in the caption. Perhaps Figure 12-15 could be eliminated. Figure 18 left and right charts seem identical.

Experimental design

The paper reports on a very technical aspect of crop type classification using remote sensing data and I am not totally convinced that it is within the scope of PeerJ. It does not deal with environmental issues relating crop type mapping and e.g. crop management, which would be in line with the scope of the journal.
This research paper could represent a case study but does not bring elements of leading edge innovation that would fill a knowledge gap. There is existing research on the topic as recent literature shows. Moreover, there are some element of criticism in the experimental design. First of all, the authors completely ignore the presence of Sentinel-2 data that has totally changed the data availability scenario and therefore the foundation of the research question and methodological approach. Of more technical relevance, the construction of the reference dataset presents some issues. The distribution of samples amongst the different classes is very uneven (e.g. cotton vs watermelon, 54 vs 7) and limited to four crop types only. There might be also a strong correlation within the ground data. This is because the training/validation samples are taken from the same fields/parcels (as shown in Figure 4c) and used as independent samples. This probably introduces a positive bias in your measures of accuracies.

Validity of the findings

The impact and novelty of the study was not totally assessed and exhaustively discussed in the context of existing studies and datasets. Ground data have limited representativeness being sampled in a limited spatial region and for a limited number of crops (4).
Results and discussion sections could be merged. Many elements of discussion are presented already with the results.
Results and conclusions are sometime presented in a contradictory way and do not answer the key question: is daily data better that aggregated data? I could not appreciate a relevant difference.
The authors report on the possibility to perform “early crop classification”, but also that (line 442) “The features of the green-up stage (Day 60–80) and the NDVI peaks (around Day 210) were the optimal features for identifying the summer crops.” From your chart in figure 5, I read that “Day 210” represent the end of season. Is this correct and how this relates to an early crop classification?
At line 445, the authors mention that “Daily NDVI time series outperformed the other composition strategies because the daily data saturated early (158-day time series)”. The OA accuracy in Table 4 does not show differences (90.23% vs 89.66%) amongst the 4 compositing strategies. When considering daily and composited data, the saturation length of the daily dataset (158-day) is essentially the same as for the 32-day compositing strategy (198-day). Isn´t it? In a real-case scenario, you do not need to wait the end of the compositing period to perform the classification and therefore the daily dataset will provide you a temporal advantage.

Additional comments

The topic of the paper is of interest but it is not fully in the scope of the Journal. I would recommend a major revision and re-submission to a different journal focusing on technical and methodological aspects.

---

## Round 0.2 · Minor Revisions

Thank you for submitting the revised version of your article and the helpful summary of the response to the reviews. The article seems to have been enhanced but there are a few minor revisions necessary. All are focussed on aspects of presentation/text quality:

1. The abstract inappropriately includes abbreviations. For example, HJ, OA, PA and UA. These are meaningless to many readers and need to be replaced (write out the meaning in full).

2. In some places subjective phrasing is used. As just one example, on line 175 reference is made to 'better' classification accuracy. Please replace with a more suitable expression such as higher or larger.

3. In some places the text is rather awkward. For example, in addressing the important issue of inter-sensor differences in spectral wavebands (text around line 180) the text is awkward. It includes odd and uncertain statements such as 'radiation inconsistency'. Please revise. Need to state clearly what the issue is and what you did about it.

Please note that the above are just examples - the whole text should be checked and polished. Addressing these minor issues will greatly enhance the clarity of the article.

---

## Round 0.3 · accepted · Accept

The revisions made seem to have enhanced the article. I am happy to Accept it

#